# Factors associated with oral health care behavior of people with type 2 diabetes mellitus: A hospital-based, cross-sectional study

Kamonchanok Sairat[1,2], Nitikorn Phoosuwan🄳[1,3]*

1 Faculty of Public Health, Kasetsart University Chalermphrakiat Sakon Nakhon Province Campus, Sakon Nakhon Province, Thailand, 2 Sub-District Health Promotion Hospitals in Bueng Khong Long, Buengkan Province, Thailand, 3 Department of Public Health and Caring Sciences, Faculty of Medicine, Uppsala University, Uppsala, Sweden

* nitikorn.ph@ku.th

## Abstract

### Background

Oral healthcare behavior determines oral health status and the incidence of oral diseases. People with type 2 diabetes mellitus (T2DM) are at-risk of having low oral healthcare behavior and disease-related oral health.

### Objective

To investigate the oral health status and factors associated with oral healthcare behavior among people with T2DM in Thailand.

### Methods

In total, 401 people with T2DM participated in the study based on their attendance at a non-communicable disease clinic at sub-district health promotion hospitals in Bueng Kan, a north-eastern province in Thailand. A structured questionnaire was used to obtain variables of interest. Linear regression analysis at the 95% confidence interval (CI) was applied.

### Results

The majority of participants were female (73.8%). More than three-quarters had more than 20 permanent teeth (77.6%), a decay missing filling tooth index was 10.6 teeth/person. Many participants had four permanent occlusal pairs (69.6%), had tooth decay (74.6%), and some participants had tooth filling (32.2%). Statistically significant factors associated with oral healthcare behavior were: having complications associated with diabetes mellitus (Beta = -0.097, 95%CI = -1.653, -0.046), oral health literacy (Beta = 0.119, 95%CI = 0.009, 0.150), educational level (Beta = 0.123, 95%CI = 0.103, 0.949), oral healthcare attitude (Beta = 0.258, 95%CI = 0.143, 0.333), and oral health services (Beta = 0.430, 95%CI = 0.298, 1.408).

**Data Availability Statement:** All relevant data are within the paper and its Supporting Information files.

**Funding:** The author(s) received no specific funding for this work.

**Competing interests:** The authors have declared that no competing interests exist.

## Conclusions

People with T2DM had good oral health status. People with T2DM with low oral health literacy, low attitude, and low level of oral health services were at a higher risk of poor oral healthcare behavior.

## Introduction

Diabetes is a chronic disease caused by a disorder in which the body produces insufficient insulin hormone. The related high blood sugar levels result in complications, such as retinopathy, kidney failure, and coronary arteries, and in the easier formation of wounds that are slow-healing, numbness of the feet, and oral disease [1]. Worldwide, more than 450 million people (9%) suffer from diabetes, which is estimated to grow more than 780 million people in 2045 [2]. In Southeast Asia, the number of people with type 2 diabetes mellitus (T2DM) is expected to increase from 90 million cases in 2021 to 152 million in the next 25 years [2].

Thailand has been classified as a middle-income country. Currently, the incidence of diabetes is continuously increasing, with about 300,000 new cases per year and about 5.3 million people with T2DM. The prevalence rate of people with T2DM was about 10% in 2014 (8.9% for men and 10.8% for women), with the rate being higher among those with a low level of education [3]. Patients with poor glycemic control are associated with oral diseases, such as tooth decay, gingivitis, oral disease, and periodontitis. These diseases result in tooth loss, a poor digestive system, which directly impact nutrient intake. People with T2DM are more likely to have an oral disease than those without T2DM [4]. Biomarkers producing immune response (e.g. transforming growth factor-β1 (TGF-β1), N-terminal portion of the B-type natriuretic propeptide, and vascular endothelial growth factor (VEGF)) play a crucial role as a mediator for immune response, result in periodontitis, especially people with chronic diseases, like diabetes [5, 6].

According to the 8th National Survey of oral health, Thai people aged 35–45 years an average of 3.6% had lost a tooth, with many of the people (62.4%) having either gingivitis problems or periodontitis, while one-quarter (25.9%) had tooth decay [7]. This oral health status can lead to tooth loss in the near future.

The northeast region in Thailand has the highest proportion of people with T2DM in the country, accounting for 38.84 percent and this is expected to increase in the coming years [8]. In Health Region 8, the incident rates of diabetes per 100,000 population for the period 2018, 2019, 2020, and 2021 were 511.53, 525.28, 580.10, and 538.65, respectively. In Bueng Kan province, Thailand, the rates of new diabetes cases per 100,000 population for the same period were 572.53, 528.48, 614.26, and 555.40, respectively, and the prevalence rate of people with T2DM was 5573.9 per 100,000 population. A screening protocol for oral health for people with T2DM aged 45–59 years showed that 19.43%, 17.15%, and 11.21% of the people were screened in the fiscal years 2020, 2021, and 2022, respectively. Of the above, 16.03%, 15.84%, and 12.11%, respectively, of the people received oral health examinations [9].

Oral healthcare behavior includes personal oral behavior, food consumption behavior, and medical care behavior [10]. Sociodemographic factors associated with oral healthcare behavior are age, sex, and educational level [11]. Knowledge factors regarding oral healthcare behavior are the core determinants of oral healthcare [12]. Attitudes, beliefs, knowledge, and behavior are also indicated, with the level of oral healthcare being interrelated with the level of knowledge of individuals [13, 14]. Furthermore, living and working conditions are related to oral

healthcare behavior; a working population with a higher level of income determines the need for more oral services [15]. However, there has been no published investigation of the oral health status, and the factors associated with oral healthcare behavior in Thailand, which is a serious omission. Thus, people with T2DM should be educated to carry out correct oral health-care, especially those with T2DM, to encourage good oral health. Hence, this study aimed to explore the factors associated with oral healthcare behavior among people with T2DM in Bueng Kan province, Thailand.

## Methods

### Aim

This hospital-based cross-sectional study was aimed at exploring oral health status and associated factors with oral healthcare behavior among people with T2DM.

### Participants

This study was carried out in Bueng Kan, a north-eastern province of Thailand located 754 kilometers from Bangkok, the capital city of Thailand. The province is close to the Lao People's Democratic Republic and the Mekong River, with a population of approximately 420,000 [9]. The province has a mixture of residential areas (rural, and semi-urban) and of cultures (Thai and Laotian). There are eight districts and 61 Sub-district Health Promotion Hospitals (SHPHs) in Bueng Kan that are responsible for approximately 4,472 people with T2DM aged 40–59 years who visit a Non-communicable Disease (NCD) clinic annually [16].

The sample size was calculated based on a formula [17], where $\alpha$ = 95%, acceptable margin of error (precision) = 0.0692, standard deviation (SD) = 0.63 [18], resulting in a minimum sample number of 351. To allow for some missing data, this study collected data from 401 people with T2DM. The participants were selected using a multi-stage random sampling method (stage one: cluster sampling by selecting the Bueng Khong Long district; stage two: stratified random sampling by a proportion of people with T2DM among males and females in each of the four SHPHs in the Bueng Khong Long district; and stage three: systematic random sampling using a population list in each SHPH sorted by name from which the samples were randomly selected from equal intervals to obtain the final samples). Bueng Khong Long district was randomly selected for data collection because people with T2DM in that district had similar characteristics (occupation, income, and education level) compared to the seven other districts. The inclusion criteria were people with T2DM who: (1) were diagnosed with T2DM by a physician for at least one year; (2) aged 40–59 years in the year of data collection; (3) received at least one of oral services (i.e. oral examination, tooth scaling, tooth extraction, tooth filling, fluoride vanishing, oral cancer screening) within one year at a SHPH in Bueng Khong Long district; (4) could communicate in Thai; and (5) had ability to provide informed consent to participate in the study. The participants were excluded if they: (1) reported having mental health problems from medical records (e.g. substance-induced psychosis); and (2) had serious complications (end stage of kidney disease).

### Instrument

A questionnaire containing seven parts was used for data collection, where the respective parts covered: (I) sociodemographic and oral healthcare of participants, (II) oral health literacy, (III) attitude to oral healthcare, (IV) oral cleaning equipment used, (V) oral health services, (VI) multi-dimensional scale of perceived social support (r-T-MSPSS), and (VII) oral healthcare behavior. The questionnaire was tested for validity by four Thai experts with PhD

qualifications and experience in healthcare services (Structure Content Validity Index: = 0.97). A reliability test was carried out using 30 people with T2DM who did not participate in the study, with the overall Cronbach's alpha coefficient = 0.708.

**Part I.** This part had 18 items and was constructed by the researchers. It consisted of questions about gender, age, marital status, occupation, education, income per month, personal expenses per month, duration of diabetes, smoking, alcohol drinking, treatment, complications, and the oral health care of the participants.

**Part II.** This part had 14 items and was created by the researchers using the health literacy questionnaire created by the Health Education Division, Ministry of Public Health, Thailand [19]. It had seven dimensions related to oral health care: (1) receptivity of oral health care, (2) understanding of oral health care, (3) social support, (4) economic barriers, (5) access to oral health services, (6) communication skills, and (7) utilization of dentistry services. Each dimension had two questions and each question had five options (0 = never done, 1 = can do, 2 = sometimes possible, 3 = frequently done, and 4 = do it regularly). Therefore, the total score of this part was in the range 0–56. This part had a Cronbach's alpha coefficient of 0.706.

**Part III.** This part consisted of eight items and was constructed by the researchers, where each question had five options (5 = strongly agree, 4 = agree, 3 = undecided, 2 = disagree,1 = strongly disagree). Therefore, each question had a score in the range 1–5 and the total possible score for this part was in the range 1–40. The reliability test was approved with a Cronbach's alpha coefficient of 0.712.

**Part IV.** This part comprised eight items and was constructed by the researchers. Each item asked about a piece of equipment that the partner did or did not use. A response of having the equipment was scored as one and of not having the equipment was scored as zero. The score for this part was in the range 0–8. The Cronbach's alpha coefficient for this part was 0.710.

**Part V.** This part comprised six items and was constructed by the researchers, where each question had three options (2 = always, 1 = sometimes, 0 = never). The score for each item was in the range 0–2 and the total possible score for this part was in the range 0–12. The reliability test for this part was satisfied (Cronbach's alpha coefficient = 0.722).

**Part VI.** This part comprised 12 items and was constructed based on the r-T-MSPSS [19], where each question had seven options (7 = very strongly agree, 6 = strongly agree, 5 = mildly agree, 4 = neutral, 3 = mildly disagree, 2 = strongly disagree, and 1 = very strongly disagree). The score for each item was in the range 1–7 and the total possible score for this part was in the range 1–84. The reliability test for this part was satisfied (Cronbach's alpha coefficient = 0.764).

**Part VII.** This part comprised 15 items that assessed the frequency of oral healthcare behavior [10]. The oral healthcare behavior covered: brush toothing behavior (eight questions), food consumption behavior (four questions), and medication behavior (three question), where each question had four options (3 = always, 2 = often, 1 = sometimes, 0 = never). The score for each item was in the range 0–3 and the total possible score for this part was in the range 0–45. The reliability test for this part was satisfied (Cronbach's alpha coefficient = 0.718).

## Ethical approval

The study was approved by the Ethics Committee, Kasetsart University, Chalermphrakiat Sakhon Nakhon province campus (ID: KU.CSC-COA65/009). In addition, permission was sought for data collection from the Bueng Khong Long District Public Health Office. All participants received written and oral information before signing a written consent form. The

information emphasized the freedom to partake in the study or not and that participants were fully entitled to withdraw at any time. None of the authors was involved in the care of any of the participants.

## Data collection

The directors of the selected SHPHs approved the data collection plan. Thereafter, the researchers received a list of people with T2DM from a nurse-midwife in the SHPHs from which the samples were randomly selected and invited to participate in the study voluntarily and informed about the study before signing a consent form. Each participant answered the questionnaire for about 15–20 minutes in a room at the SHPH where they received oral health services. The researchers obtained oral health information about the participants from the medical records of the SHPH, such as oral health status (number of permanent teeth, number of occlusal pairs, tooth decay, teeth missing, teeth filling). Data were collected from the 1st of May to 30th of September 2022.

## Statistical analyses

Data were analyzed using a statistical software program. Descriptive statistics (such as frequency and mean) were calculated. Inferential statistics were conducted using regression analysis, while the dependent variable was the score for oral healthcare behavior.

The independent variables were based on socio-demographics and oral health care of participants, oral health literacy score, score of attitudes to oral health care, oral cleaning equipment score, score for oral health services, and a score for the r-T-MSPSS. The independent and dependent variables were verified for the assumptions of the linear regression analysis based on a test of normality for the dependent variable ($Z_{skewness}$ = -0.435, $Z_{Kurtosis}$ = 0.372), a test for multicollinearity problems, and the variance inflation factor.

All independent variables were included in univariable analysis for the first step. Only significant independent variables ($p < 0.05$) from the first step were retained in the multivariable analysis based on the enter method. The degree of association was assessed using the 95% confidence interval (CI), the 0.05 significance level was used, and the coefficient of determination ($R^2$) was used to describe the relationship between predictors and the oral healthcare behavior of people with T2DM.

## Results

In total, data from 401 out of 445 people with T2DM were analyzed (response rate 90.1%). Overall, majority of the study participants were females (73.8%), 50–59 years old (72/1%), married (79.3%), agriculturists (59.4%), graduated in primary education level (83.8%), non-smokers (92.0%), non-alcohol drinkers (84.3%). See Table 1.

Majority of the study participants were people with T2DM diagnosed for 1–5 years (44.4%), used oral drug treatment (89.3%), had permanent teeth more than 20 (77.6%), had occlusal pairs more than 4 (69.6%), and the Decay Missing Filled Tooth (DMFT) index was 10.6 teeth/person. See Table 2.

The majority of the people with T2DM had the highest means of support (S1) and were able to have an accompanying family member or friend to an oral appointment (mean = 3.51, SD = 0.73), followed by receptivity (R1) and being able to pay attention to oral or oral health needs (mean = 3.31, SD = 0.73), with the T2DM group having the lowest mean of understanding (U2) and able to read oral or oral health information brochures left in oral clinics and waiting rooms. See Table 3.

**Table 1. Characteristics of the participants in the study (n = 401).**

| Characteristic | Frequency | Percentage |
|---|---:|---:|
| Gender | | |
| Female | 296 | 73.8 |
| Male | 105 | 26.2 |
| Age group (years) | | |
| 40–49 | 112 | 27.9 |
| 50–59 | 289 | 72.1 |
| Mean (SD) = 52.39 (4.94) Min = 40 Max = 59 | | |
| Marital Status | | |
| Married | 318 | 79.3 |
| Single | 38 | 9.5 |
| Divorced | 36 | 9.0 |
| Separated | 7 | 1.7 |
| Widowed | 2 | 0.5 |
| Employment | | |
| Agriculture | 238 | 59.4 |
| Unemployed | 82 | 20.4 |
| General employment | 34 | 8.5 |
| Personal business | 21 | 5.2 |
| Trade | 10 | 2.5 |
| Others (e.g. government service) | 16 | 4.0 |
| Educational level | | |
| Primary | 336 | 83.8 |
| Lower Secondary | 26 | 6.5 |
| Upper Secondary | 28 | 7.0 |
| Diploma | 6 | 1.5 |
| Bachelor's degree | 5 | 1.2 |
| Income per month (USD) | | |
| 0–175 | 246 | 61.3 |
| 176–350 | 95 | 23.7 |
| More than 350 | 60 | 15.0 |
| Mean (SD) = 218.61 (218.26) Min = 14.6 Max = 1456.8 | | |
| Personal expenses per month (USD) | | |
| 0–175 | 288 | 71.8 |
| 176–350 | 83 | 20.7 |
| More than 350 | 30 | 7.5 |
| Mean (SD) = 157.23 (161.37) Min = 5.83 Max = 1165.5 | | |
| Smoking | | |
| No | 369 | 92.0 |
| Yes | 25 | 6.2 |
| Ex- | 7 | 1.8 |
| Alcohol drinker | | |
| No | 338 | 84.3 |
| Yes | 55 | 13.7 |
| Ex- | 8 | 2.0 |

Note: USD 1 = THB 34.32; SD = standard deviation.

**Table 2. General health and oral health status of the participants in the study (n = 401).**

| General health | Frequency | Percentage |
|---|---|---|
| Duration of diabetes | | |
| 1–5 years | 178 | 44.4 |
| 6–10 years | 142 | 35.4 |
| More than 10 | 81 | 20.2 |
| Mean (SD) = 7.84 (5.47) Min = 1.0 Max = 30.0 | | |
| Treatment (multiple answers allowed) | | |
| Oral drugs | 358 | 89.3 |
| Diet | 191 | 47.6 |
| Insulin | 16 | 4.0 |
| Combinations of oral drugs and insulin | 16 | 4.0 |
| Complications (multiple answers allowed) | | |
| No | 355 | 88.5 |
| Yes | 46 | 11.5 |
| Eyes | 29 | 7.2 |
| Feet | 22 | 5.5 |
| Kidneys | 8 | 2.0 |
| Hearts | 5 | 1.3 |
| **Oral health status** | **Frequency** | **Percentage** |
| Number of permanent teeth | | |
| ≥ 20 | 311 | 77.6 |
| < 20Mean (SD) = 24.00 (6.97) Min = 0.0 Max = 32.0 | 90 | 22.4 |
| Number of occlusal pairs | | |
| ≥ 4 | 279 | 69.6 |
| < 4 | 122 | 30.4 |
| Tooth Decay | | |
| Yes | 299 | 74.6 |
| No | 102 | 25.4 |
| Mean (SD) = 1.97 (1.48) Min = 0.0 Max = 10.0 | | |
| Tooth Missing | | |
| Yes | 348 | 86.8 |
| No | 53 | 13.2 |
| Mean (SD) = 7.87 (6.86) Min = 0.0 Max = 32.0 | | |
| Tooth Filling | | |
| No | 272 | 67.8 |
| Yes | 129 | 32.2 |
| Mean (SD) = 1.76 (1.35) Min = 0.0 Max = 6.0 | | |
| Decay Missing Filling Tooth index | | |
| Mean (SD) = 10.60 (7.03) Min = 0.0 Max = 32.0 | | |

Note: SD = standard deviation

The findings revealed that having the complications of diabetes mellitus (Beta = -0.097, 95%CI = -1.653, -0.046), oral health literacy (Beta = 0.119, 95%CI = 0.009, 0.150), educational level (Beta = 0.123, 95%CI = 0.103, 0.949), oral health care attitude (Beta = 0.258, 95% CI = 0.143, 0.333), and oral health services (Beta = 0.430, 95%CI = 0.298, 1.408) were statistically significant factors associated with the oral healthcare behavior of people with T2DM. The variables were able to predict 33.0% of the oral healthcare behavior of people with T2DM ($R^2$ = 0.330). See Table 4.

**Table 3. Oral health literacy of study participants (n = 401).**

| Oral health literacy domains | Oral health literacy items | Mean Score (SD) |
|---|---|---|
| Receptivity (R) | (R1) Are you able to pay attention to your oral or oral health needs?<br>(R2) Are you able to make time for things that are good for your oral or oral health? | 3.31 (0.73)<br>3.32 (0.72) |
| Understanding (U) | (U1) Are you able to fill out oral forms such as the enrollment form?<br>(U2) Are you able to read oral or oral health information brochures left in oral clinics and waiting rooms? | 3.10 (1.01)<br>2.99 (1.07) |
| Support (S) | (S1) Are you able to take a family member or friend with you to an oral appointment?<br>(S2) Are you able to ask a family member or friend for help to understand oral or oral health information? | 3.51 (0.73)<br>3.36 (0.75) |
| Economic barriers (E) | (E1) Are you able to pay to see a dentist?<br>(E2) Are you able to afford transportation to the oral clinic? | 3.18 (0.77)<br>3.36 (0.75) |
| Access (A) | (A1) Do you know where a dentist can be contacted?<br>(A2) Do you know what to do to get a dentist's appointment? | 3.20 (0.75)<br>3.20 (0.75) |
| Communication (C) | (C1) Are you able to look for a second opinion about your oral health from a oral health professional?<br>(C2) Are you able to use information from a dentist to make decisions about your oral health? | 3.09 (0.78)<br>3.19(0.81) |
| Utilization (X) | (X1) Are you able to carry out instructions that a dentist gives you?<br>(X2) Are you able to use advice from a dentist to make decisions about your oral health? | 3.37 (0.70)<br>3.30 (0.68) |

Note: SD = standard deviation

## Discussion

This study focused on providing a synthesis of current evidence on the oral health services, oral health care attitudes, and oral health literacy of people with T2DM and the association of these factors with oral healthcare behavior. A large proportion of respondents rated their oral health status as tooth decay (74.6%) with an average of 7.87% having lost a tooth, which was higher than the Thai national survey of oral health in Thailand (tooth decay = 49.7% and 3.2% having lost a tooth) [20]. Having complications of diabetes mellitus, low oral health literacy, low education, low oral health care attitude, low access of oral health services were factors associated with the low oral healthcare behavior of people with T2DM.

In the northeastern region, 17.4% of working-age groups and 19.4% of elderly people reportedly visited oral checkup services without symptoms, which were lower proportions

**Table 4. Factors associated with oral healthcare behavior of people with T2DM based on multivariable linear regression analysis (n = 401).**

| Variables | Unstandardized coefficients | | Standardized coefficients | t | p-value | 95% Confidence for Interval for B | |
|---|---|---|---|---|---|---|---|
| | b | Std. Error | Beta | | | Lower Bound | Upper Bound |
| Complications of diabetes mellitus | -0.1424 | 0.625 | -0.097 | -2.276 | 0.023* | -1.653 | -0.046 |
| Oral health literacy | 0.079 | 0.036 | 0.119 | 2.205 | 0.028* | 0.009 | 0.150 |
| Educational level | 0.753 | 0.268 | 0.123 | 2.815 | 0.005* | 0.103 | 0.949 |
| Oral health care attitude | 0.238 | 0.048 | 0.258 | 4.913 | <0.001* | 0.143 | 0.333 |
| Oral health services | 1.403 | 0.155 | 0.430 | 9.036 | <0.001* | 0.298 | 1.408 |
| (Constant) | 4.254 | 2.238 | - | 1.901 | 0.058 | -0.147 | 8.653 |

$R = .57$ $R^2 = .330$ $F = 19.194$

*Correlation significant at p-value < 0.05.

than overall of those seeking services for treatment [21]. People with T2DM lack access to oral services; hence, the introduction of the tele-oral system plays a role in screening. Assessment of the risk of oral diseases and early detection of diseases, such as screening for oral caries and gingivitis and examination of oral lesions by sending information to a specialist dentist for diagnosis and receiving advice, leads to the organization of disease promotion and prevention programs according to the patient's risk [22]. The provision of proactive oral services to promote oral health is required to prevent the occurrence or progression of oral diseases [23]. Periodontal disease is more severe in people with T2DM than in normal people and it causes bad breath and loose teeth, resulting in tooth loss and a decreased ability to chew [24]. Furthermore, receiving oral health care by scaling reduces blood sugar levels (HbA1c) by approximately 0.27–1.03% [25]. Uncontrolled blood sugar levels lead to oral disease. Risky behaviors, e.g. high sugar intake, smoking, alcohol drinking, and low physical activities, may cause oxidative stress reactions; these result in periodontal inflammation and severity of T2DM, and affect to quality of life of the people with T2DM. [5, 6]. Therefore, people with T2DM should receive oral services at least once a year to effectively reduce the risk of periodontitis [26].

Oral health literacy could contribute to increased positive oral health behavior among the participants in this study. According to the World Health Organization [27], "health literacy is a person's cognitive and social skills that cause motivation and competence to reach understand and use health information and services to promote and maintain good health"; promoting oral health literacy is an economical way for developing countries to achieve better oral health outcomes [28]. Oral health care information access skills include mobile phone and computer applications to access information more easily, such as Facebook [29]. Channels for accessing oral health information and knowledge about oral hygiene care are necessary as a supplement to preventing oral caries and periodontal disease for the public [20], with the level of basic health literacy stemming from access to reliable and correct information. This includes understanding dentistry and having correct oral health care knowledge [30]. Therefore, oral health literacy (skills in accessing oral health information, understanding oral health care, and exchanging knowledge) might create the awareness and interest of people [31], who subsequently may make a decision to improve their oral health behavior in a sustainability manner.

## Strengths and limitations

In this study, multivariable linear regression analysis was used to adjust for confounders and to demonstrate the strengths of the association. Although this study was restricted to studying people in a government setting, most people with T2DM in Thailand still use services from government sources. Thus, the findings can be referred to other people with T2DM in similar contexts and cultures, such as agricultural areas in the Lao People's Democratic Republic. On the other hand, the research design of this study used a cross-sectional study, which might limit the causal relationship between the findings. A case-control study might be conducted in the future to reduce biases.

## Conclusions

The value of the present study is its provision of new information regarding oral healthcare behavior, oral health status, and oral diseases. The majority of participants (people with type 2 diabetes mellitus) were female (73.8%), and 77.6% of the participants had more than 20 permanent teeth and 69.6% had four permanent occlusal pairs. The decay missing filled tooth (DMFT) index was 10.6 teeth/person. However, about 75% of the participants had tooth decay (74.6%) and more than 30% of the participants had tooth filling. Complications from diabetes mellitus, educational level, oral health care attitude, and oral health services were statistically

significant factors associated with oral healthcare behavior. People with T2DM who have low oral health literacy, low attitude, and a low level of oral health services were at a higher risk of poor oral healthcare behavior and are in need of an intervention program, such as programs focusing on food consumption behavior medication behavior.

## Supporting information

**S1 Checklist. STROBE statement—checklist of items that should be included in reports of observational studies.**
(DOCX)

**S1 Dataset.**
(XLS)

**S1 File. Ethical approval.**
(JPG)

## Acknowledgments

The authors thank all the respondents who participated in this study, the Faculty of Public Health, Kasetsart University for providing resources and materials, and the government Sub-district health promotion hospitals in Bueng Khong Long, Buengkan Province, Thailand, Thailand for their cooperation.

## Author Contributions

**Conceptualization:** Kamonchanok Sairat, Nitikorn Phoosuwan.

**Data curation:** Kamonchanok Sairat.

**Formal analysis:** Kamonchanok Sairat, Nitikorn Phoosuwan.

**Investigation:** Kamonchanok Sairat, Nitikorn Phoosuwan.

**Methodology:** Kamonchanok Sairat, Nitikorn Phoosuwan.

**Project administration:** Kamonchanok Sairat, Nitikorn Phoosuwan.

**Resources:** Nitikorn Phoosuwan.

**Supervision:** Kamonchanok Sairat, Nitikorn Phoosuwan.

**Writing – original draft:** Kamonchanok Sairat.

**Writing – review & editing:** Kamonchanok Sairat, Nitikorn Phoosuwan.

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
