## [Decision Letter · Decision Letter 0]

11 Mar 2024

PONE-D-23-32407Factors associated with oral health care behavior of people with type 2 diabetes mellitus: A hospital-based, cross-sectional studyPLOS ONE

Dear Dr. Phoosuwan,

Thank you for submitting your manuscript to PLOS ONE. After careful consideration, we feel that it has merit but does not fully meet PLOS ONE’s publication criteria as it currently stands. Therefore, we invite you to submit a revised version of the manuscript that addresses the points raised during the review process.

We look forward to receiving your revised manuscript.

Kind regards,

Gaetano Isola, Ph.D.

Academic Editor

PLOS ONE

Journal Requirements:

Reviewers' comments:

Reviewer's Responses to Questions

**Comments to the Author**

1. Is the manuscript technically sound, and do the data support the conclusions?

Reviewer #1: Partly

Reviewer #2: Yes

Reviewer #3: Yes

2. Has the statistical analysis been performed appropriately and rigorously? 

Reviewer #1: Yes

Reviewer #2: Yes

Reviewer #3: Yes

3. Have the authors made all data underlying the findings in their manuscript fully available?

Reviewer #1: Yes

Reviewer #2: No

Reviewer #3: Yes

4. Is the manuscript presented in an intelligible fashion and written in standard English?

Reviewer #1: Yes

Reviewer #2: No

Reviewer #3: No

5. Review Comments to the Author

Reviewer #1: In the manuscript entitled: "Factors associated with oral health care behavior of people with type 2 diabetes mellitus: A hospital-based, cross-sectional study" the authors aimed to assess the factors associated with oral healthcare behavior among people with T2DM in Thailand.

The authors found that the majority of participants were female (73.8%) and most of the participants

had received primary education (83.8%). More than three-quarters (77.6%) had more than 20 permanent teeth, while many of the participants (69.6%) had four permanent occlusal pairs. A decay missing filling tooth index was 10.6 teeth/person.

The authors concluded that people with T2DM had good oral health status. People with T2DM with

low oral health literacy, low attitude, and low level of oral health services were at a higher risk of poor oral healthcare behavior.

Major comments:

In general, the idea and innovation of this study regards the analysis of oral health in patients with diabetes is interesting and novel because the role these aspects in medicine are validated but further studies on this topic could be an innovative issue in this field could be open a creative matter of debate in literature by adding new information. Moreover, there are few reports in the literature that studied this interesting topic with this kind of study design.

The study was well conducted by the authors; However, there are some concerns to revise that are described below.

The introduction section resumes the existing knowledge regarding the important factor linked with the impact mediators involved together oral health and with periodontitis.

However, as the importance of the topic, the reviewer strongly recommends, before a further re-evaluation of the manuscript, to update the literature through read, discuss and must cites in the references with great attention all of those recent interesting articles, that helps the authors to better introduce and discuss the role of mediators (NT-PRO-BNP and TGF beta 1) in periodontitis and related recessions by adding as a references these article, before any further assessment of the manuscript: 1) DOI: doi: 10.1002/JPER.23-0063. PMID: 37433155; 2) DOI: 10.1177/1721727X1301100217

The authors should be better specified, at the end of the introduction section, the rationale of the study and the aim of the study. In the central section, should better clarify inclusions and exclusions criteria of the selected sample.

Please better state the results obtained in the abstract.

The discussion section appears well organized with the relevant paper that support the conclusions, even if the authors should better discuss the relationship regarding the by periodontitis in and risk of oxidative stress evolution that could improve the quality of life in periodontitis patients which undergo diabetes. The conclusion should reinforce in light of the discussions.

In conclusion, I am sure that the authors are fine clinicians who achieve very nice results with their adopted protocol. However, this study, in my view does not in its current form satisfy a very high scientific requirement for publication in this journal and requests a revision before a futher re-evaluation of the manuscript.

Minor Comments:

Abstract:

- Better formulate the abstract section by better describing the aim of the study

Introduction:

- Please refer to major comments

Discussion

- Please add a specific sentence that clarifies the results obtained in the first part of the discussion

Reviewer #2: Manuscript Number: PONE-D-23-32407

Full Title: Factors associated with oral health care behaviour of people with T2DM

Dear Authors.

Overall I think it is a good article that needs a few changes or additions (See below). I would suggest accepting with minor revisions.

Methods

1. Authors should clearly define what they mean by 'oral healthcare behavior.' (e.g The actions and habits individuals adopt....).

2. Authors should mention and describe their cross-sectional design method somewhere in the Methods, preferably at the start (e.g., in a design section).

Discussion

1. Authors should consider providing suggestions for future research based on gaps they identified, and their limitations.

Conclusions

1. The authors should also expand on what they mean by 'intervention program'.

Reviewer #3: A relevant and important study as it relates to oral health and systemic disease (DM) especially among LMIC and MMIC. will certainly add to the current scarcity of information on these issues in the Asian environment particularly.

Methodology well described. Concern is on data presentation and interpretation of Results especially Table 1. subsequently needs to look into refinement of discussion. Conclusion acceptable. actually Abstract is better written

6. PLOS authors have the option to publish the peer review history of their article (what does this mean?). If published, this will include your full peer review and any attached files.

Reviewer #1: No

Reviewer #2: **Yes: **Wondwossen Fantaye Abawollo

Reviewer #3: No

---

## [Author Response · Author response to Decision Letter 0]

27 Mar 2024

Response to reviewers' comments

Dear Editor,

We thank you and the reviewers for remarks which have helped us improve the manuscript. Please see our response below.

Reviewer #1: In the manuscript entitled: "Factors associated with oral health care behavior of people with type 2 diabetes mellitus: A hospital-based, cross-sectional study" the authors aimed to assess the factors associated with oral healthcare behavior among people with T2DM in Thailand. The authors found that the majority of participants were female (73.8%) and most of the participants had received primary education (83.8%). More than three-quarters (77.6%) had more than 20 permanent teeth, while many of the participants (69.6%) had four permanent occlusal pairs. A decay missing filling tooth index was 10.6 teeth/person. The authors concluded that people with T2DM had good oral health status. People with T2DM with low oral health literacy, low attitude, and low level of oral health services were at a higher risk of poor oral healthcare behavior.

Major comments: In general, the idea and innovation of this study regards the analysis of oral health in patients with diabetes is interesting and novel because the role these aspects in medicine are validated but further studies on this topic could be an innovative issue in this field could be open a creative matter of debate in literature by adding new information. Moreover, there are few reports in the literature that studied this interesting topic with this kind of study design. The study was well conducted by the authors; However, there are some concerns to revise that are described below.

The introduction section resumes the existing knowledge regarding the important factor linked with the impact mediators involved together oral health and with periodontitis. However, as the importance of the topic, the reviewer strongly recommends, before a further re-evaluation of the manuscript, to update the literature through read, discuss and must cites in the references with great attention all of those recent interesting articles, that helps the authors to better introduce and discuss the role of mediators (NT-PRO-BNP and TGF beta 1) in periodontitis and related recessions by adding as a references these article, before any further assessment of the manuscript: 1) DOI: doi: 10.1002/JPER.23-0063. PMID: 37433155; 2) DOI: 10.1177/1721727X1301100217

Response: Thank you for your suggestion and comments. We have read the articles and written texts in the Introduction section, page 3/25 and added in the reference list.

The authors should be better specified, at the end of the introduction section, the rationale of the study and the aim of the study. 

Response: We have already specified the rationale and aim of the study in the introduction section, page 4/25 and in the Aim section page 5/25.

In the central section, should better clarify inclusions and exclusions criteria of the selected sample.

Response: We have already clarified the criteria in the Participants section, page 6/25.

Please better state the results obtained in the abstract.

Response: We have already rewritten and added texts for the results in the Results section, pages 9-10/25.

The discussion section appears well organized with the relevant paper that support the conclusions, even if the authors should better discuss the relationship regarding the by periodontitis in and risk of oxidative stress evolution that could improve the quality of life in periodontitis patients which undergo diabetes. 

Response: We have already rewritten and added texts in the Discussion section, page 12/25.

The conclusion should reinforce in light of the discussions.

Response: We have already rewritten and added texts in the Conclusions section, page 13/25.

Minor Comments:

Abstract:

- Better formulate the abstract section by better describing the aim of the study

Response: We have already rewritten the aim of the study in the Abstract section, page 2/25.

Discussion

- Please add a specific sentence that clarifies the results obtained in the first part of the discussion

Response: We have already added texts in the Discussion section, page11/25.

Reviewer #2: Manuscript Number: PONE-D-23-32407

Full Title: Factors associated with oral health care behaviour of people with T2DM 

Dear Authors. Overall, I think it is a good article that needs a few changes or additions (See below). I would suggest accepting with minor revisions.

Methods

1. Authors should clearly define what they mean by 'oral healthcare behavior.' (e.g. the actions and habits individuals adopt....).

Response: We have already added texts to clarify this in the Introduction section, page 4/25.

2. Authors should mention and describe their cross-sectional design method somewhere in the Methods, preferably at the start (e.g., in a design section).

Response: We have already added texts for the design of the study in the Methods section, page 5/25.

Discussion

1. Authors should consider providing suggestions for future research based on gaps they identified, and their limitations.

Response: We have already added texts to suggest for future research in the Strengths and limitations section, page 13/25.

Conclusions

1. The authors should also expand on what they mean by 'intervention program'.

Response: We have already added texts in the Conclusions section, page 13/25.

Reviewer #3: A relevant and important study as it relates to oral health and systemic disease (DM) especially among LMIC and MMIC. will certainly add to the current scarcity of information on these issues in the Asian environment particularly. Methodology well described. Concern is on data presentation and interpretation of Results especially Table 1. subsequently needs to look into refinement of discussion. Conclusion acceptable. Actually Abstract is better written.

Response: Thank you for your comments. We have rewritten results accordingly in the Results section, pages 9-10/25 and in the Abstract section page 2/25.

---

## [Decision Letter · Decision Letter 1]

26 Apr 2024

Factors associated with oral health care behavior of people with type 2 diabetes mellitus: A hospital-based, cross-sectional study

PONE-D-23-32407R1

Dear Dr. Phoosuwan,

We’re pleased to inform you that your manuscript has been judged scientifically suitable for publication and will be formally accepted for publication once it meets all outstanding technical requirements.

Kind regards,

Gaetano Isola, Ph.D.

Academic Editor

PLOS ONE

Additional Editor Comments (optional):

The manuscript can be accepted for publication.

Reviewers' comments:

Reviewer's Responses to Questions

**Comments to the Author**

1. If the authors have adequately addressed your comments raised in a previous round of review and you feel that this manuscript is now acceptable for publication, you may indicate that here to bypass the “Comments to the Author” section, enter your conflict of interest statement in the “Confidential to Editor” section, and submit your "Accept" recommendation.

Reviewer #1: All comments have been addressed

Reviewer #3: (No Response)

2. Is the manuscript technically sound, and do the data support the conclusions?

Reviewer #1: Yes

Reviewer #3: Yes

3. Has the statistical analysis been performed appropriately and rigorously? 

Reviewer #1: Yes

Reviewer #3: Yes

4. Have the authors made all data underlying the findings in their manuscript fully available?

Reviewer #1: Yes

Reviewer #3: Yes

5. Is the manuscript presented in an intelligible fashion and written in standard English?

Reviewer #1: Yes

Reviewer #3: Yes

6. Review Comments to the Author

Reviewer #1: In this revised version of the manuscript, the authors have well addressed to all of the concerns raised by the reviewer. The manuscript can be accepted for publication.

Reviewer #3: Keywords . Behavior; Factor standing alone is confusing. Perhaps better to join as Behavioral Factors; Oral healthcare ; etc.

Participants and Results - your original sample after randomization is actually 445 as stated in Row 24 Results Section first line. This was not mentioned earlier in Participants section. Agreed you analyzed on 401 but calculating 401 (90.1%) response is not correct. Better to say, data was analyzed on a final sample of 401 after considering inclusion-exclusion criteria at the end of Participant section or at Row 24 Results.

Index DMF used to measure caries - F not filling but FILLED

there's a few grammatical error needing attention

7. PLOS authors have the option to publish the peer review history of their article (what does this mean?). If published, this will include your full peer review and any attached files.

Reviewer #1: No

Reviewer #3: No

---

## [Editor Report · Acceptance letter]

8 May 2024

PONE-D-23-32407R1 

PLOS ONE

Dear Dr. Phoosuwan, 

I'm pleased to inform you that your manuscript has been deemed suitable for publication in PLOS ONE. Congratulations! Your manuscript is now being handed over to our production team.

Kind regards, 

on behalf of

Prof. Gaetano Isola 

Academic Editor

PLOS ONE